# Deep Learning for Caries Detection and Classification

**DOI:** 10.3390/diagnostics11091672

**Published:** 2021-09-13

**Authors:** Luya Lian, Tianer Zhu, Fudong Zhu, Haihua Zhu

**Affiliations:** Stomatology Hospital, School of Stomatology, Zhejiang University School of Medicine, Clinical Research Center for Oral Diseases of Zhejiang Province, Key Laboratory of Oral Biomedical Research of Zhejiang Province, Cancer Center of Zhejiang University, Hangzhou 310006, China; 3100102358@zju.edu.cn (L.L.); zhutianer@zju.edu.cn (T.Z.)

**Keywords:** deep learning methods, caries diagnosis, dental panoramic images, radiography

## Abstract

Objectives: Deep learning methods have achieved impressive diagnostic performance in the field of radiology. The current study aimed to use deep learning methods to detect caries lesions, classify different radiographic extensions on panoramic films, and compare the classification results with those of expert dentists. Methods: A total of 1160 dental panoramic films were evaluated by three expert dentists. All caries lesions in the films were marked with circles, whose combination was defined as the reference dataset. A training and validation dataset (1071) and a test dataset (89) were then established from the reference dataset. A convolutional neural network, called nnU-Net, was applied to detect caries lesions, and DenseNet121 was applied to classify the lesions according to their depths (dentin lesions in the outer, middle, or inner third D1/2/3 of dentin). The performance of the test dataset in the trained nnU-Net and DenseNet121 models was compared with the results of six expert dentists in terms of the intersection over union (IoU), Dice coefficient, accuracy, precision, recall, negative predictive value (NPV), and F1-score metrics. Results: nnU-Net yielded caries lesion segmentation IoU and Dice coefficient values of 0.785 and 0.663, respectively, and the accuracy and recall rate of nnU-Net were 0.986 and 0.821, respectively. The results of the expert dentists and the neural network were shown to be no different in terms of accuracy, precision, recall, NPV, and F1-score. For caries depth classification, DenseNet121 showed an overall accuracy of 0.957 for D1 lesions, 0.832 for D2 lesions, and 0.863 for D3 lesions. The recall results of the D1/D2/D3 lesions were 0.765, 0.652, and 0.918, respectively. All metric values, including accuracy, precision, recall, NPV, and F1-score values, were proven to be no different from those of the experienced dentists. Conclusion: In detecting and classifying caries lesions on dental panoramic radiographs, the performance of deep learning methods was similar to that of expert dentists. The impact of applying these well-trained neural networks for disease diagnosis and treatment decision making should be explored.

## 1. Introduction

Dental caries are common causes of tooth pain and tooth loss, despite being preventable and treatable. Comprehensive and early detection of dental caries can be critical for timely and appropriate treatment. Large, clearly visible tooth cavities induced by caries can be easily detected by using visual inspection and probing with the use of a dental probe and a handheld mirror. These conventional caries detection methods are also effective for partially obscured but accessible caries [1]. X-ray radiography, as an aid for the diagnosis of hidden or inaccessible lesions, is irreplaceable. Panoramic, periapical, and bitewing X-rays are three common types of radiographs that are widely used in clinical practice. Bitewing and periapical X-rays concentrate on the details of the mouth area, such as one or more teeth, whereas panoramic X-rays capture all the teeth and other hard tissues of the maxillofacial region [2]. Although bitewing radiography is the most widely used approach to detect caries lesions and assess their depth, which comes with high sensitivity and specificity [3], it could not perform comprehensive lesions detection of the full mouth in one attempt. Furthermore, panoramic films are taken outside the mouth and have better patient acceptance, a lower infection rate, and a lower radiation exposure [4]. Due to its relative cost effectiveness and diagnostic evidence, panoramic imaging is considered to be the most common and important radiological tool for clinical dental disease screening, diagnosis, and treatment evaluation.

During the diagnosis and treatment of oral diseases, dentists need to interpret panoramic radiographs and record specific symptoms of diseased teeth in the medical records. New dentists require extensive training and time to perform accurate X-ray film interpretations [5]. An X-ray analysis showed that more experienced dentists are almost four times more likely to make a correct assessment of caries lesions than less experienced dentists [6]. Therefore, considerable attention has been given to interpreting panoramic X-rays with dental caries automatically. In recent decades, scientists have tried to deploy machine learning techniques to detect dental diseases. As in the conventional method, operators or experts perform lesion detection and evaluation on radiographs manually and objectively. This task is tedious when facing large amounts of image data and may lead to misinterpretations. Previous efforts have successfully applied convolutional neural network (CNN)-based deep learning models in computer vision. Deep learning methods do not depend on well-designed manual features and have high generalization capabilities. These models have achieved high accuracy and sensitivity and are the most advanced technology for a wide range of applications. The increased interest in deep learning methods has also led to their applications in medical imaging interpretation and in diagnostic assistance systems, for instance, Helicobacter pylori infection detection in gastrointestinal endoscopy [7], skin cancer screenings [8], and coronavirus disease 2019 (COVID-19) detection in computed tomography images [9].

In dentistry, Ronneberger employed U-Net to achieve dental structure segmentation on bitewing radiographs since 2015 [10]. Subsequently, CNNs have been employed with high accuracy to detect alveolar bone loss in periapical X-rays and panoramic X-rays and to identify apical cysts and caries lesions in periapical X-rays [11]. To date, multiple deep learning methods have been used for caries detection in bitewings [12,13,14] and periapical radiographs [14,15] and other auxiliary testing images such as near-infrared light transillumination images [13,16]. Most previous studies have been limited to lesion segmentation analysis of deep learning models [12,13,14,15]. Subsequently, recent research aimed to compare the caries detection performance of deep learning methods and dentists [12,17]. However, there are few studies on neural networks’ performance of caries lesions with different radiographic depths. The latter is of great importance to health economic perspectives and treatment decision making, since dental caries treatments, such as remineralization, cavity filling, root canal therapy, and tooth extraction, vary with lesion depth. As for this purpose, Cantus applied U-Net to classify caries depth on 3686 bitewing radiographs and concluded that a deep neural network was more accurate than dentists when detecting caries on bitewing radiographs [12]. However, no study has yet investigated caries lesions segmentation along with classification on panoramic films, which are of great importance in caries screening and diagnosis in primary hospitals. A previous study suggested that dentinal involvement, indicating operative treatment, had a cutoff value of 3 according to a modified International Caries Detection and Assessment System (ICDAS II). For all ICDAS II, the relative dentinal depth of a lesion was expressed as the percentage of the total length of the coronal dentin in histological and radiographic assessments. We focused on dentinal carious decay and divided the entire caries depth into four levels.

In this study, to achieve accurate segmentation of dental caries and diagnosis of lesion extensions, we used nnU-Net and DenseNet121. First, we applied nnU-Net to perform caries lesion segmentation. This segmentation model was based on a deep learning method and inspired by the structure of U-Net, which allowed us to optimally configure the model. This feature allows the model to perform outstandingly in any new task [10]. Second, we proposed DenseNet121 to identify caries stages. This 121-layer connected network alleviated the vanishing gradient issue and strengthened feature propagation by joining all proceeding layers into subsequent layers [18]. Finally, to ensure that the structure attains the best possible performance, we added a dropout mechanism and label softening to the model to address the overfitting phenomenon during model training. Moreover, we compared the caries detection results of dentists and the model to search for a better way to clinically diagnose caries lesions.

Accordingly, the main contributions of our study are threefold: (1) we built a new dataset that was strictly verified by dental experts, (2) we addressed automatic caries lesion segmentation by nnU-Net and applied DenseNet121 to automatically clarify lesion extensions into four levels, (3) we also compared the results of our model with those of a group of experienced dentists to confirm the hypothesis that a combined panoramic interpretation by the model and dentists is more sufficient and accurate than separate interpretations by a dentist or by the model.

## 2. Materials and Methods

### 2.1. Study Design

In the current study, the performances of a group of individual dentists and two deep learning methods in identifying caries lesions and their extensions in panoramic images were compared in different dimensions. This study followed the guidelines of the Standards for Reporting of Diagnostic Accuracy Studies (STARD) [19].

Before the study, our group successfully performed automated tooth segmentation, which is the cornerstone of automated diagnostic methodologies for dental films. In the present study, we first applied nnU-Net, which is well known for its state-of-the-art performance on 23 public datasets; nnU-Net is a deep learning-based segmentation method that has been broadly used for medical imaging segmentation tasks and has been proven to surpass countless prevailing approaches without manual intervention [20].

Second, we used the DenseNet121 classification model to identify carious lesions with different degrees of severity, which were previously labeled by three independent dentists. DenseNet [18] was proposed by Huang to solve the vanishing gradient problem of CNN structures, and its performance exceeded the best performance of ResNet in 2016. The key concept of DenseNet is the “skip connection”, and it has a CNN structure with dense connections. In this network, all preceding layers’ outputs are combined and input into the next layer. Moreover, to prevent losing information during layer-to-layer transmission and to overcome the vanishing gradient problem, the feature map learned by the exact layer is directly transmitted to all the following layers as output. With this model, each pixel that belongs to a radiograph can be distributed into a propriate class; in our study, there were the following four classes: “D0” sound; “D1” caries radiolucency in enamel or in the outer third of dentin; “D2” caries radiolucency in the middle third of dentin; and “D3” caries radiolucency in the inner third of dentin with or without apparent pulp involvement (Table 1).

To evaluate the performance of trained models, it is necessary to define metrics in the automated approach to measure the level of congruency between the predicted regions and the truly affected regions. Intersection over union (IoU) was the first metric that we leveraged in the present study. It is a widely used parameter that measures the difference between the ground truth region and the predicted region, as it calculates the ratio of the intersection and union of the two areas. To be more accurate, the Dice coefficient was applied to focus on the overlap of the predicted region with the ground truth region to obtain pixel accuracy. To focus on medical significance, other metrics (mainly at the tooth level) were adopted in the current study and are described below.

### 2.2. Reference Dataset

A set of 1160 panoramic images that originated from dental treatments and routine care were provided by the Affiliated Stomatology Hospital, Zhejiang University School of Medicine. A representative sample was drawn from 2015 and 2020. Panoramic images and metadata, i.e., sex, age, and image creation date, were available. However, the metadata were only allowed for descriptive analyses. The data collection process of the study was ethically approved by the Chinese Stomatological Association ethics committee. Only panoramic images of permanent teeth were included, and those of primary teeth or blurred images were excluded. The mean age (SD, min–max) of the patients included in the dataset was 42.8 (15.3, 18–68) years. Approximately 58% of the patients were male, and 42% of the patients were female. The radiographic data were all generated with radiographic machines from Dentsply Sirona (Bensheim, Germany), Orthophos XG 5OS Ceph.

Three dental experts independently labeled the images in triplicate by using the annotation tool itksnap. Each annotation was further classified into four stages according to the caries lesion depth in the radiographic films by three independent dentists. No clinical records were obtained or assessed in the procedure. For a single image, a consensus of the expert dentists was required to determine the final label, i.e., the experts were asked to repeatedly evaluate caries extensions regarding different opinions, and then, a fourth expert reviewed and revised all of the labels, including addition, deletion, and confirmation operations. All expert dentists were employed at the Affiliated Stomatology Hospital, Zhejiang University School of Medicine and had clinical experience of 3–15 years. A handbook that indicated how to mark caries lesions and annotate their stages with an annotation tool was used to guide the experts. All annotated areas on an image ultimately constructed the reference dataset (the “ground truth”), which consists of 1166 D1 lesions, 1039 D2 lesions, and 1635 D3 lesions.

### 2.3. Segmentation and Classification Model

The deep learning model applied in dental caries segmentation is nnU-Net, which is different from other improved U-Net-based models. It automatically configures itself, including preprocessing, network architecture, training, and postprocessing, for any new task, to achieve the best performance. The nnU-Net automated method configuration begins with extracting the dataset fingerprint and then executing heuristic rules. A set of fixed parameters, empirical decisions, and interdependent rules are modeled in this process [20]. Similar to other U-Net-derived architectures, a U-shaped configuration of convolutional network layers with skip connections is designed. The network architecture consists of an encoder (the falling part of the “U”) and a corresponding decoder (the rising part of the “U”). The encoder network increases the contextual information, condenses the input sequence, and decreases the exact positional information. With the skip connection between the falling and the rising part of the “U”, the decoder network expands the contextual information and combines it with precise information about the object locations [21]. The details of the model architecture are provided in Figure 1.

The DenseNet model is a CNN and is applied in caries classification. All features used in the previous layers of the architecture are reused in the current layer, and this heavy feature reuse characteristic in each block makes the network focus on efficiency. Due to this structure, the number of parameters in the DenseNet model is reduced, and the feature maps are significantly smaller, as the number of feature maps increases linearly with the growth rate. Moreover, compression layers are applied between dense blocks to keep the feature map sizes small. In addition, the network uses bottlenecks to reduce the number of parameters and the computational effort [18]. The details of the model architecture and the implementation details are presented in Figure 2.

The model was implemented by using Ubuntu (version 18.04), pytorch1.6, CUDA10.1, and CUDNN8.0.5.

### 2.4. Model Training and Data Preparation

#### 2.4.1. Data Preparation

According to caries labeling, a 300 × 400 region-of-interest (ROI) image for each caries area was cut from the panoramic radiographs to form a caries classification dataset. Then, the classification data were divided into the training set and the test set. Horizontal flip, vertical flip, horizontal vertical flip, and random rotation data enhancement operations were adopted for the training set data, and the rotation angle was within 0 and 15 degrees.

#### 2.4.2. Model Training

DenseNet121 was proposed to identify caries lesion extensions. To overcome the small size of the dataset, we used transfer learning during model training. The introduction of transfer learning is reported to save computation time and resources and enable a rapid convergence for the model. To use transfer learning, the pretrained DenseNet121 network transfers parameters to the target DenseNet121 model, which prevents overfitting. We first trained DenseNet121 on the ImageNet dataset and then used the caries dataset to fine-tune the pretrained DenseNet121 to complete caries extension classification.

Overfitting is a common problem that occurs when a CNN with a large number of learnable parameters is trained on a relatively small dataset. As the learned weights are designed mostly for the training set and lack the ability to be generalized to unseen data, the model is prone to obtaining poor performance on the test data not included in the training set. The overfitting problem is believed to be caused by the complex coadaptation of neurons, which is why deep neural networks depend on their joint response rather than favoring each neuron to perform valuable feature learning [22]. Imposing a stochastic behavior in the forward data propagation phase of the network is a commonly used method to enhance the generalization ability of CNNs [23]. Examples of such methods include label smoothing and dropout. We choose dropout [24] to randomly shut down some features and enhance the model’s generalization ability; moreover, each time before the activation function is applied, batch normalization is applied to further improve the effect. Label smoothing is another simple but successful regularization approach applied in the study. This method is widely used for multiclass classification tasks, where the CE error is adopted as the standard loss function, and the so-called one-hot encoding is presented in an annotation format. Label smoothing is designed to replace hard labels with smoothed versions; furthermore, label smoothing can prevent overconfident models when calculating the loss value and has been reported to increase the learning speed and benefit the overall accuracy [25]. Label smoothing has been proven to improve model calibration and out-of-distribution detection [26]. Label softening is equivalent to reducing the weight of the category of the real sample label when calculating the loss function and finally has the effect of suppressing overfitting.

### 2.5. Comparator Dentists

A group of six dentists who worked at Affiliated Stomatology Hospital, Zhejiang University School of Medicine, for 3–15 years were defined as the comparable group. They were enlisted to gauge the performance of the expert dentists against the performance of the neural networks. Each of the participants performed caries segmentation and severity classification tasks on a set of 89 panoramic films (test dataset), which included films of 40 D1 lesions, 53 D2 lesions, and 103 D3 lesions and images without lesions.

### 2.6. Evaluation Metrics

#### 2.6.1. Performance of nnU-Net in Caries Segmentation

The nnU-Net segmentation model was evaluated, and its performance was compared with that of the doctors. Two distinct metrics, the IoU and the Dice coefficient metrics, were used to evaluate the performance of different dimensions. Despite the similarity of the two metrics, a single instance of bad segmentation was penalized much more in the IoU than in the Dice coefficient. For certain algorithms, the vast majority of instances are correct, but incorrect decisions are made in a few instances. The model Dice coefficient score will be much higher than the corresponding IoU score, which means that the Dice coefficient reflects the average performance better and is not overly sensitive to a few bad results. Both the IoU and the Dice coefficient are calculated by the mean value in a performance assessment. The Dice coefficients indicate the mean value of individual Dice coefficients on the validation and test data. Dice coefficient and IoU values of 1 indicate an ideal algorithm that matches the reference labels 100%. In contrast, the reference and predicted label masks with no overlap will result in two metric values equal to 0.

#### 2.6.2. Performance of DenseNet121 in the Classification of Caries Severity

DenseNet121 was applied for caries severity classification, its performance was evaluated and compared with that of dentists combined with a neural network, and the precision of both was evaluated at the caries level. An ensemble of six different metrics was deployed to capture different aspects of the classification performance of the model and the dentists, including accuracy, recall, specificity, precision, F1-score, and negative predicted value (NPV). The F1-score parameter is the harmonic average of precision and recall. The chi-square test was used to compare the performances of the model and the dentists. A *p*-value with *p* < 0.05 was considered significant.

## 3. Results

Table 2 shows the distribution of caries lesions and their extensions in the reference dataset. The image ratio of the training set versus the test set was 982:89. Table 3 shows the segmentation performances of nnU-Net and of the dentists in the test set. Table 4 summarizes the performances of DenseNet121 and of the dentists in stratifying lesions to different extensions in the test set.

First, the binary classification results of nnU-Net and the dentists are presented. The overall accuracy of the model was 0.986, and the mean accuracy of the dentists was lower than that of the model but not significantly at 0.955 (min–max: 0.933–0.972; CI: 95%; *p* > 0.05). The IoU scores of the model and the dentists were 0.785 and 0.696 (min–max: 0.711–0.717; CI: 95%; *p* > 0.05), respectively. The Dice coefficient scores of the model and the dentists were 0.663 and 0.570 (min–max: 0.587–0.594; CI: 95%; *p* > 0.05), respectively. The model yielded a better accuracy, precision, recall, specificity, NPV, F1-score, IoU, and Dice scores than the dentists, while the results of all metrics showed no significant difference between the model and the dentists (CI: 95%; *p* > 0.05)).

Second, we considered multiclass classification for DenseNet121 and analyzed the performances of the dentists and of deep learning methods for dental caries stage diagnosis. For D1 lesions, the recall rate of the model was 0.765, while it was 0.466 for the dentists (CI: 95%; *p* > 0.05). For D2 lesions, the recall rate of the model was 0.652, while it was 0.539 for the dentists (CI: 95%; *p* > 0.05). For D3 lesions, the recall rate of the model was 0.918, while it was 0.954 for the dentists (CI: 95%; *p* > 0.05). Although there were no significant differences between the sensitivity scores of the dentists and those of the model for all caries stages, the model seemed to be more sensitive in detecting D1 and D2 lesions. The same results were found for accuracy, specificity, precision, NPV, and F1-score metrics. Even though no significant differences were found in the previous metrics, the model yielded higher scores in terms of all metrics for D1 and D2 lesions than the dentists. The recall, NPV, and F1-score values of the dentists for D3 lesions were slightly higher than those of the model.

## 4. Discussion

Due to the varying accuracy and sensitivity of individual dentists in the detection of caries lesions and their depth, inconsistent treatment decisions and suboptimal care are quite common. High-throughput diagnostic assistance provided by computer-assisted analysis tools could support dentists with these procedures. To date, panoramic films, as the main auxiliary diagnostic method for oral disease screening, have been gradually interpreted by deep learning. However, deep learning has very rarely been used in caries depth classification. Furthermore, the performance of these models is not regularly compared with that of dentists in caries lesion segmentation or classification [27]. The latter (lesion stage-specific classification performance) is of vital importance in clinical decision making. Enamel caries can be treated by remineralization, and dentin caries in the outer space are commonly treated by cavity filling. For deep dentin caries that approach the dental pulp, pulp capping or root canal therapy is required. From this perspective, the automatic and accurate panoramic interpretation of dental caries lesion staging can provide comprehensive treatment recommendations for individuals. This study aimed to design an intelligence-assisted diagnosis method based on a combined nnU-net and DenseNet121 model to replace the manual interpretation of caries lesions and their extensions. We achieved these goals by constructing caries panoramic datasets for four-stage caries extensions. Furthermore, the segmentation performance of nnU-net and the classification performance of DenseNet121 were evaluated individually and in combination with dentist diagnoses to carry out a comparative analysis.

Our results suggest that nnU-Net can be used for the automated interpretation of panoramas to facilitate caries diagnosis. The accuracy of the model was higher than that of models in previous studies [12,28] and yielded a score of 0.986. The performances of the model and of the experienced dentists showed no significant difference in caries lesion segmentation. However, nnU-Net seems to be more efficient and achieved reliable and objective results.

In our study, DenseNet121 proved to be effective in lesion extension classification. Combining transfer learning with simplified image preprocessing improved the classification accuracy and recall of the neural network. It is prudent to conclude that this method allows us to automatically learn the differences among the caries types in caries extension image features and attain valid interpretations.

The results indicate that the model that we used in the study can automatically learn the differences among caries depths in caries extension image features and achieve effective interpretations. Although the chi-square tests of accuracy, recall, specificity, precision, NPV, F1-score, IoU, and Dice metrics between the model and the dentists showed no significant differences (CI: 95%; *p* > 0.5), the model yielded better scores than the dentists for D1 and D2 lesions. Moreover, the model seemed to be more efficacious and reliable than the dentists, since the six experienced dentists did not show good consistency and stability, the 95% confidence interval for the ICC population values of the dentists was 0.595 (0.537 < ICC < 0.653), and the model was much faster and more accurate in lesion classification. Further research needs to be conducted with a larger dataset and different experienced dentists.

In this study, DenseNet121 seemed to be more sensitive in classifying D1 and D2 lesions and had similar recall rates when compared to dentists in classifying D3 lesions, which is consistent with our hypothesis. Notably, in clinical radiograph interpretation, D3 lesions have a larger range of transmission images in panoramic films and are easier to detect with the naked eye. Caries in the D1 and D2 stages are more likely to be missed or have lesion boundaries that are difficult to determine. However, the recall rates of the dentists and the model were not significantly different according to the chi-squared test. The result was as expected. The dentists involved for the comparison were all experienced experts, and their results were used to set the “ground truth”. However, larger tests involving more dentists from different departments and with different experience levels may obtain different results in further studies. For better performance, a combination of dentists’ diagnoses and the model’s results to detect caries and perform classification is recommended.

Nevertheless, it is challenging to achieve satisfying segmentation results due to the slight difference in the gray levels between tooth structures and bone on panoramic films [29]. Complicated changes in the pixel intensity of overlapping skeletal structures in panoramic films are a particular obstacle to overcome. These structures include the nasal area, maxillary sinus, teeth, and surrounding bone [30]. Moreover, our targets (caries lesions) are quite small when compared to the whole image. For these reasons, we enlarged the radiographs to 1:5 when labeling. However, some boundaries of the lesions were undefinable in overlapping two-dimensional images. Moreover, we have constantly increased the dataset and have now built a dataset with 3840 caries verified by experts.

This study has some strengths and limitations. First, we built a large dataset relative to other datasets in the dental field. Since there is no open dataset related to caries stages in relevant research fields, 1160 panoramic X-rays were meticulously collected, and blurred images were excluded. Three expert dentists were trained to label and annotate the dental caries, and a fourth expert revised any controversial results. Second, the predicted caries were output as highlighted areas by nnU-Net and presented in three different colors according to their depths obtained by DenseNet121. Third, the aforementioned performance comparison between dental experts and nnU-Net and DenseNet121 was carried out on a test dataset. As a limitation, our panoramic films were made on the equipment of one company and we excluded the blurred ones (90 out of 1250) before training the models, which means that the reference dataset underlying our research is not fully generalizable. It is essential to verify our neural networks on an external test set in the next steps. Furthermore, we applied no gold standard in the study such as micro-CT and histology of extracted teeth. However, dentists with different experiences and professional backgrounds are required for comparison, which may provide more valuable information. Second, labeling in the constructed reference test was not sufficiently precise, as it was not triangulated with the gold standard (histology). Even without a hard gold standard, “fuzzy” labeling should be verified with data from other diagnostic approaches, such as visual, tactile, or transillumination inspection, if possible. Finally, nnU-Net and DenseNet121 have not been executed or implemented in an auxiliary diagnosis system until now. It is difficult to infer whether the model will have a positive impact when it is actually deployed in patient care [31].

Accordingly, we recommend that further studies use well-trained neural networks in random and prospective designs. The accuracy of neural networks and the correct usage of these tools in the clinic should be explored. This correct usage includes how dentists adopt and interact with the tools, how the diagnostic procedure improves, and how the tools change the treatment decision-making protocol. Before entering clinical care, all deep learning methods are recommended to be reviewed according to the standards of evidence-based practice, and then, a comprehensive set of results should be obtained in various environments to ensure their robustness, universality, and clinical consequences.

## 5. Conclusions

Accordingly, the well-trained neural network performed similarly to experienced dentists in detecting caries lesions and classifying them according to depth within our limited study. Notably, although the dentists and the neural network seemed to have a similar performance, the neural network might have better sensitivity and accuracy in classifying caries extensions in the outer dentin. The impact of using the network on the accurate diagnosis of diseases and treatment decision making should be further explored.

## Figures and Tables

**Figure 1 diagnostics-11-01672-f001:**
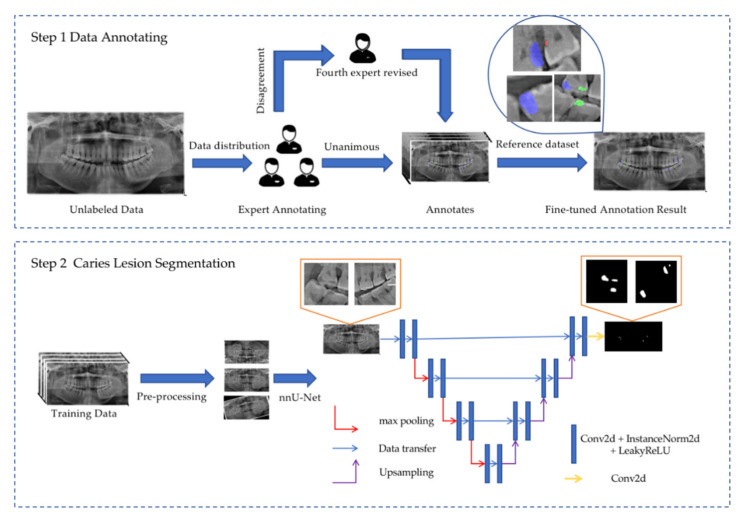
Details of nnU-Net architecture and implementation details in caries segmentation. In step 1, three dental experts were trained to implement dental caries labels and annotations, and a fourth expert revised any controversial results. Purple circle indicates D3 lesions, green circle indicates D2 lesions and red circle indicates D1 lesions. Step 2 shows nnU-Net and how it works in caries lesion segmentation.

**Figure 2 diagnostics-11-01672-f002:**
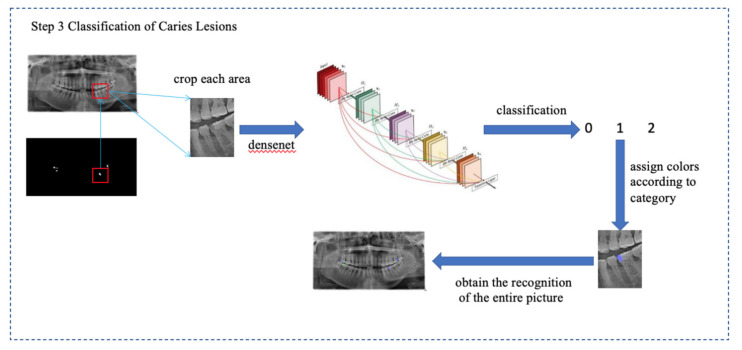
Description of how caries lesions were classified into D1/D2/D3 lesions. First, caries lesions were identified by the model as 3 types, which were represented by the code 0/1/2. Code 0 indicates D1 lesions (which are shown as red circles), Code 1 indicates D2 lesions (which are shown as green circles) and Code 3 indicates D3 lesions (which are shown as purple circl.

**Table 1 diagnostics-11-01672-t001:** Criterion of caries extension and their stage.

Caries Stage	Caries Extension
D0	Sound
D1	Caries radiolucency in enamel or in the outer third of dentin
D2	Caries radiolucency in the middle third of dentin
D3	Caries radiolucency in the inner third of dentin with or without apparent pulp involvement

**Table 2 diagnostics-11-01672-t002:** Reference dataset.

Dataset	D1	D2	D3
Training set	1126	986	1532
Test set	40	53	103
Overall	1166	1039	1635

**Table 3 diagnostics-11-01672-t003:** Segmentation performances of nnU-Net and the dentists with the test set.

	Accuracy	Sensitivity	Specificity	Precision	NPV	F1	IoU	Dice
Model	0.986	0.821	1.000	1.000	0.985	0.902	0.785	0.663
Dentists (mean)	0.955	0.773	0.971	0.705	0.981	0.733	0.696	0.570
Dentists (min)	0.933	0.730	0.949	0.554	0.977	0.632	0.711	0.587
Dentists (max)	0.972	0.852	0.992	0.883	0.987	0.802	0.717	0.594

**Table 4 diagnostics-11-01672-t004:** Classification performances of DenseNet121 and the dentists with the test set.

Parameter	DenseNet121			Dentists (Mean; Min–Max)		
	D1	D2	D3	D1	D2	D3
Accuracy	0.957	0.832	0.863	0.915; 0.886–0.940	0.792; 0.720–0.828	0.858; 0.783–0.903
Precision	0.812	0.732	0.865	0.798; 0.667–1.000	0.601; 0.458–0.677	0.847; 0.737–0.884
Sensitivity	0.765	0.652	0.918	0.464; 0.250–0.647	0.536; 0.290–0.630	0.947; 0.881–0.988
NPV ^1^	0.972	0.867	0.860	0.926; 0.891–0.956	0.847; 0.773–0.878	0.895; 0.745–0.966
F1-score	0.788	0.690	0.891	0.570; 0.400–0.645	0.564; 0.355–0.630	0.892; 0,844–0.929

^1^ NPV: negative predictive value. Please see the main text for the definitions of the metrics.

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
