# Peer review of "Deep Learning for Caries Detection and Classification"

_diagnostics, 2021, doi:10.3390/diagnostics11091672_

Round 1
Reviewer 1 Report
Dear authors, the topic is very interesting and it represents the new future for helping a clinician to make diagnosis and planning. Even if a panoramic is not the gold standard to detect caries detection this software can give a great help to the clinician during the practice. A minor issue: you excluded from the study blurred images; can you precise how was the percentage of these images? Can you also insert this exclusion as a limitation of the study?Author Response
Please see the attachment.

Reviewer 2 Report
This is an interesting paper regarding the use of deep learning for the detection of dental caries in panoramic films.
The manuscript should be carefully checked for grammatical issues.
There have been several recent papers describing the use of deep learning for caries detection in dental radiographs and other imaging technologies. The authors citation of prior work is quite limited. The authors should summarize the prior work and describe the advantages and disadvantages of each imaging method and the performance of prior deep learning approaches for caries detection.
In the authors approach no gold standard was employed. A group of expert dentists was utilized to present the “ground truth”. The authors argue that they have demonstrated that their algorithms can perform as well as expert dentists. The expert dentists had a clinical experience of 3-15 years which is not that extensive. I don’t see dentists running out to purchase software that performs as well as an expert dentist. I would think a more valuable study would utilize deep learning to demonstrate a significant improvement in diagnostic accuracy using some sort of gold standard since it is well known that radiographs have low sensitivity and typically underestimate the true lesion depth. Why not perform a study in which histology or microCT data is available on extracted teeth ? Please describe why a gold standard was not used in this study.
Reviewer 3 Report
I accept manuscript to publish after minor revision.
Dear Authors.
Congratulations on your work which, I found interesting.
Manuscript: Deep Learning for Caries Detection and Classification, it is well written with an adequate structure as a scientific paper demands.
I have some minor revisions to propose to you to improve your work. Please refer to the following comments:
- The panoramic pictures are not the image of choice for caries diagnosis. It is advisable to take bitewing pictures. It is worth giving this information in the introduction.
- The study was based on panoramic images made on the equipment of one company. It is not known how the program dealt with images from various devices. This is mentioned in the materials and methods, but it is worth repeating in the discussion regarding the limitations of the study.
Round 2
Reviewer 2 Report
Authors have suitably addressed my concerns